# Ceftiofur formulation differentially affects the intestinal drug concentration, resistance of fecal *Escherichia coli*, and the microbiome of steers

Derek M. Foster[1]*, Megan E. Jacob[1], Kyle A. Farmer[1], Benjamin J. Callahan[1], Casey M. Theriot[1], Sophia Kathariou[2], Natalia Cernicchiaro[3], Timo Prange[4], Mark G. Papich[5]

**1** Department of Population Health and Pathobiology, College of Veterinary Medicine, NC State University, Raleigh, NC, United States of America, **2** Department of Food, Bioprocessing, and Nutrition Sciences, College of Agriculture and Life Sciences, NC State University, Raleigh, NC, United States of America, **3** Department of Diagnostic Medicine and Pathobiology, College of Veterinary Medicine, Kansas State University, Manhattan, KS, United States of America, **4** Department of Clinical Sciences, College of Veterinary Medicine, NC State University, Raleigh, NC, United States of America, **5** Department of Molecular and Biomedical Sciences, College of Veterinary Medicine, NC State University, Raleigh, NC, United States of America

* derek_foster@ncsu.edu

**Data Availability Statement:** All sequence files are available from the Bioproject database (ID PRJNA560079).

## Abstract

Antimicrobial drug concentrations in the gastrointestinal tract likely drive antimicrobial resistance in enteric bacteria. Our objective was to determine the concentration of ceftiofur and its metabolites in the gastrointestinal tract of steers treated with ceftiofur crystalline-free acid (CCFA) or ceftiofur hydrochloride (CHCL), determine the effect of these drugs on the minimum inhibitory concentration (MIC) of fecal *Escherichia coli*, and evaluate shifts in the microbiome. Steers were administered either a single dose (6.6 mg/kg) of CCFA or 2.2 mg/kg of CHCL every 24 hours for 3 days. Ceftiofur and its metabolites were measured in the plasma, interstitium, ileum and colon. The concentration and MIC of fecal *E. coli* and the fecal microbiota composition were assessed after treatment. The maximum concentration of ceftiofur was higher in all sampled locations of steers treated with CHCL. Measurable drug persisted longer in the intestine of CCFA-treated steers. There was a significant decrease in *E. coli* concentration (P = 0.002) within 24 hours that persisted for 2 weeks after CCFA treatment. In CHCL-treated steers, the mean MIC of ceftiofur in *E. coli* peaked at 48 hours (mean MIC = 20.45 ug/ml, 95% CI = 10.29–40.63 ug/ml), and in CCFA-treated steers, mean MIC peaked at 96 hours (mean MIC = 10.68 ug/ml, 95% CI = 5.47–20.85 ug/ml). Shifts in the microbiome of steers in both groups were due to reductions in Firmicutes and increases in Bacteroidetes. CCFA leads to prolonged, low intestinal drug concentrations, and is associated with decreased *E. coli* concentration, an increased MIC of ceftiofur in *E. coli* at specific time points, and shifts in the fecal microbiota. CHCL led to higher intestinal drug concentrations over a shorter duration. Effects on *E. coli* concentration and the microbiome were smaller in this group, but the increase in the MIC of ceftiofur in fecal *E. coli* was similar.

**Funding:** DMF: This work is/was supported by the USDA National Institute of Food and Agriculture, project 1010130. The funders had no role in study design, data collection and analysis, decision to publish, or preparation of the manuscript.

**Competing interests:** DMF has received research support from Zoetis. MGP has received gifts, honoraria, consulting fees, and research support from Zoetis, the manufacturer of ceftiofur. This does not alter our adherence to PLOS ONE policies on sharing data and materials.

## Introduction

Ceftiofur, a third generation cephalosporin, is one of the most common antimicrobials administered to feedlot cattle and lactating dairy cows in the United States for treatment of respiratory disease, [1,2] metritis, [3] and is also used in an extralabel manner to treat enteric disease [4]. This use has led to widespread concern over selection for bacteria with antimicrobial resistance (AMR) in the feces of treated cattle that could be transferred to humans through the food chain [5]. To determine this risk, studies have determined the presence of AMR genes in dairy cattle feces immediately after ceftiofur treatment [6], tested the susceptibility of *E. coli* isolates from preweaned calves and cows on farms that use ceftiofur [7,8], and quantified *bla (CMY-2)* and/or *bla(CTX-M)* in feedlot steers [9–12] and dairy cattle [13] treated with ceftiofur. Due to the variability in the populations studied and the outcome measures, the conclusions of these studies have varied widely, precluding any clear recommendations for prudent use of ceftiofur in cattle.

Further complicating the association between ceftiofur use and AMR outcomes is the availability of different drug formulations, which varied across previous studies. Ceftiofur hydrochloride (CHCL) is a 50 mg/ml oil-based suspension that is FDA-approved for daily administration for 3–5 days at 1.1–2.2 mg/kg by subcutaneous or intramuscular route. Cattle cannot be slaughtered for human consumption within 4 days of the last treatment with CHCL (Zoetis). Ceftiofur crystalline-free acid (CCFA) is a 200 mg/ml oil-based suspension that is administered as single-dose therapy at 6.6 mg/kg with a meat withdrawal time of 14 days. Due to the slow-release formulation, CCFA must be administered subcutaneously on the posterior aspect of the ear or at the base of the ear (Zoetis). Some producers may use CHCL because of the shorter withdrawal time and easier route of administration, while others may prefer CCFA due to the ease of single-dose therapy. Although both products are FDA-approved for similar conditions in cattle and used somewhat interchangeably, they produce different exposure profiles, which could affect selection of antibiotic-resistant bacteria. To our knowledge, a comparison of the effect of these formulations on AMR in fecal bacteria of treated cattle has not been performed.

We hypothesize that the concentration of antimicrobials within the gastrointestinal tract (GIT) is significantly associated with the risk of AMR in fecal bacteria as measured by an increase in MIC. Surprisingly, this association remains unknown because it has been difficult to directly obtain intestinal drug concentration data. We showed in our previous studies that continuous collection of luminal fluid from ileum and colon is possible, and allowed for measurement of drug concentrations in intestinal fluid and pharmacokinetic modelling of the active drug concentrations during the time after drug injection [14–16]. The objective of the current study was to compare the active antimicrobial concentrations in the GIT of steers treated with either CHCL or CCFA, and correlate those concentrations with changes in fecal bacteria, including changes in the minimum inhibitory concentration of ceftiofur in *E. coli*. In addition to serving as indicator organism for foodborne pathogens, *E. coli* has human health importance and can acquire relevant resistance to third-generation cephalosporins.

## Materials and methods

### Animals and treatments

This study was approved by the North Carolina State University Institutional Animal Care and Use Committee (protocol # 18-020A). This study took place May through July of 2016. Twelve six-month-old Holstein steers (186 to 288 kg) were obtained from the North Carolina State University Dairy Educational Unit as was done in previous studies [15,17–19]. Sample size was determined based on the number needed for appropriate pharmacokinetic modeling

from our previous studies in cattle [14,15,19]. Investigators were not blinded to the treatment groups of the steers. Steers were fitted for placement of ultrafiltration probes in the ileum and spiral colon as described below. At 24–48 hours post-probe placement, animals received one of two treatments (n = 6 steers per treatment)—subcutaneous injection of ceftiofur crystalline-free acid (CCFA; Excede®; 6.6 mg/kg) as a single dose at the base of the ear, or ceftiofur hydrochloride (CHCL; Excenel®; 2.2 mg/kg) subcutaneously in the neck every 24 hours for 3 treatments. Using a parallel study design, all procedures and sample collection with CCFA steers were completed first, and then treatment and sampling of CHCL steers were completed, so steers were not randomly allocated to treatment groups. Steers were housed in pairs that received the same treatment in stalls bedded with shavings and were fed grass hay with free access to water for the duration of the study. At the conclusion of the study and observation of the appropriate meat withdrawal time, all ultrafiltration probes and catheters were removed, and the steers were sold.

### Plasma collection

Prior to drug administration, a jugular catheter (Intracath®, Becton Dickinson, Franklin Lakes, NJ) was inserted into the jugular vein. Blood samples (6 ml) were collected in lithium heparin tubes at appropriate intervals for optimum pharmacokinetic modeling of each drug for at least 3 drug half-lives, accounting for 90% of drug elimination from the plasma. These were time 0, 15 min, 30 min, 1, 2, 4, 8, 12, 24, 32, 48, 72, 96, 120, 144, 168, 192 hours for CCFA and time 0, 15 min, 30 min, 1, 2, 4, 8, 12, 24, 25, 26, 28, 30, 32, 36, 48, 48.25, 48.5, 49, 50, 52, 54, 56, 60, 72, 74, 78, 96 hr after the initial dose for CHCL. The tubes were immediately centrifuged at 1,000 x g for 10 minutes to collect plasma and stored at -80˚C until assayed.

### Placement of intestinal ultrafiltration probes

Surgical procedures took place over 4 days with 3 surgeries per day as previously described [16] with the following variation in the anesthesia procedure. Briefly, food and water were withheld from all steers for 12 hours prior to surgery. The steers were restrained standing in a conventional chute. The right flank was anesthetized by infiltration of 2% lidocaine dorsally and ventrally to the lateral vertebral processes of L1-L4 (approximately 80 ml per steer). After entering the abdomen, the ileum and colon were identified by first retracting the cecum through the flank incision. The collecting loops of an ultrafiltration probe (UF-3-12, BAS; Bioanalytical Systems, West Lafayette, IN, USA) were inserted into the lumen of the ileum and spiral colon, and sutured into place. The free ends of the probes were exteriorized cranial to the skin incision. The calves received 2 mg/kg of flunixin meglumine intravenously prior to surgery and 24 hours after surgery according to the IACUC protocol. There is no evidence of any impact on ceftiofur pharmacokinetics due to flunixin administration [20].

### Gastrointestinal fluid collection

After surgery, the probes were prepared to collect samples of fluid from the ileum and spiral colon of each animal. The tubing exiting the body cavity was connected to a needle within a vacuum vial needle holder using flexible tubing and secured. The vial holder was sutured to the skin over the transverse processes of the lumbar spine using 2–0 nylon (Ethicon; Somerville, NJ) and white tape butterfly tags. To collect the ultrafiltrate, a 3-ml evacuated tube with no additive (Becton-Dickinson) was inserted onto the needle of the vacuum vial needle holder. The ultrafiltrate collected is free of protein and other intestinal contents that could potentially bind to the antibiotic. Drug administration and sample collection began 24–48 hours after surgery. Steers were allowed free-choice grass hay and water after recovery from surgery until the

completion of the study. Samples from probes placed in the ileum and spiral colon were collected 0, 2, 4, 8, 12, 24, 32, 48, 72, 96, 120, 144, 168, and 192 hours post administration of CCFA and 0, 2, 4, 6, 8, 12, 24, 26, 28, 30, 32, 36, 48, 50, 52, 54, 56, 60, 72, 74, 78, and 96 hours post administration of initial CHCL dose by changing the tubes at the predetermined time points.

### Interstitial fluid collection

An *in-vivo* ultrafiltration probe was also inserted in the subcutaneous space above the shoulders in a manner described in previous studies (Davis et al., 2007; Messenger et al., 2012). The interstitial fluid (ISF) was collected at time 0 (pre-treatment) and at the same time points as for the GI fluid collection. The collected fluid was immediately frozen at -80˚C for further analysis.

### Determining active drug concentration

Plasma and tissue fluid samples were analyzed by reverse-phase high pressure liquid chromatography (HPLC) with ultraviolet detection to determine the active concentrations of ceftiofur and its metabolites as previously described [21,22]. Ceftiofur is rapidly metabolized to the active metabolite desfuroylceftiofur in cattle, which is the predominant metabolite responsible for antibacterial effects. The assay converts all ceftiofur and desfuroylceftiofur conjugates to a single stable derivative, desfuroylceftiofur acetamide, which is measured by HPLC ultraviolet detection. All drug concentrations were determined from calibration curves made from fortified (spiked) blank plasma, intestinal and interstitial fluid collected from the experimental calves prior to antibiotic administration. Calibration curves were prepared from fortifying the blank matrix with reference drug standards of ceftiofur (United States Pharmacopeia {USP}, Rockville, MD) to validate the HPLC analysis and perform Quality Control (QC) assessments during the assay.

### Pharmacokinetic analysis

The drug concentrations were analyzed using standard pharmacokinetic methods to examine the drug disposition for each calf. A computer program (Phoenix® WinNonlin®, V. 8.0; Pharsight Corporation, Certara, St. Louis MO) was used to determine pharmacokinetic (PK) parameters.

Plasma, ISF, and intestinal drug concentrations were plotted on linear and semi-logarithmic graphs for analysis and for visual assessment of the best model for pharmacokinetic analysis. Specific models (e.g., one, two, etc. compartments) were determined on the basis of visual analysis for goodness of fit and by visual inspection of residual plots. The best model fit was based on the equation described in the following formula:

$$C = \frac{k01FD}{V(k01 - k10)}(e^{-k10t} - e^{-k01t})$$

Where C is the plasma concentration, t is time, $k_{01}$ is the non-IV absorption rate, assuming first-order absorption, $k_{10}$ is the elimination rate constant, V is the apparent volume of distribution, F is the fraction of drug absorbed, and D is the non-IV dose. Secondary parameters calculated from the model included the peak concentration ($C_{MAX}$), time to peak concentration ($T_{MAX}$), area under the plasma-concentration vs time profile (AUC), and the respective absorption and terminal half-lives (t½).

A compartmental model could not be fit to all of the concentrations from the intestinal fluid samples because of sparse sampling (incomplete collection) in some calves. Therefore, data from some calves were analyzed using noncompartmental analysis (NCA) in the same pharmacokinetic program described above. For the NCA, the area under the plasma concentration vs time curve (AUC) from time 0 to the last measured concentration, (defined by the

limit of quantification) was calculated using the log-linear trapezoidal method. The AUC from time 0 to infinity was calculated by adding the terminal portion of the curve, estimated from the relationship $C_n/\lambda_Z$, to the $AUC_{0\ Cn}$, where $\lambda_Z$ is the terminal slope of the curve, and $C_n$ is the last measured concentration point.

The relative drug transfer from the plasma compartment to the ISF and intestinal fluids was measured by calculation of a *penetration factor*. The penetration factor was determined by the ratio of AUC for the intestinal fluid to the AUC for plasma:

$$\text{Penetration Factor} = \frac{AUC_{\text{Intestinal fluid or ISF}}}{AUC_{\text{Plasma}}}$$

## Fecal sampling

Individual fecal samples ($>$50 g) from each steer were manually collected from the rectum using a clean rectal sleeve and sterile lubricant immediately prior to treatment, and at 24, 36, and 48 hours after treatment. Following this period, fecal samples were collected approximately every day through 7 days and again at 14 days after drug administration.

## Quantification of *E. coli* from feces

One gram of feces was inoculated into 9 ml EC broth (Oxoid Ltd., Basingstoke, Hampshire, England) and vortexed. One ml was immediately removed and serially diluted ten-fold in sterile phosphate-buffered saline, and 100 μl was plated in triplicate onto HardyCHROM$^{\text{TM}}$ ECC Media (Hardy Diagnostics, Santa Maria, CA). Plates were incubated overnight at 37°C and dilutions that yielded between 30 and 300 pink-violet colonies on each of the 3 plates were counted and counts were averaged to determine the concentration (CFU/g) of *E. coli* at each time point. The remaining EC broth was incubated overnight at 37°C, and if no growth was observed on direct plates, the enrichment was streaked for isolation on ECC plates and incubated overnight at 37°C. From the quantified or enrichment plate, eight colonies were randomly picked, streaked onto Columbia agar with 5% sheep blood (Remel, Lenexa, KS) and incubated overnight at 37°C. Following incubation, each isolate was transferred to a 2-ml cryogenic vial containing LB Broth (Sigma-Aldrich, St. Louis, MO) supplemented with 25% glycerol, vortexed, and frozen at -80°C.

## Determination of minimum inhibitory concentration

The minimum inhibitory concentration (MIC) of ceftiofur for each *E. coli* isolate was determined using a broth microdilution method and following Clinical and Laboratory Standards Institute standards [23]. Isolates were grown overnight on blood agar. A single colony was inoculated into 3 ml of sterile phosphate-buffered saline and brought to a 0.5 McFarland Standard, then 10 μl of this Standard were further diluted in 990 μl sterile phosphate-buffered saline. This final bacterial suspension (50 μl) was inoculated into 50 μl of two-fold serial dilutions of ceftiofur (USP, Rockville, MD) in Mueller Hinton broth (BD, Sparks, MD) ranging in concentration from 0.03 to 32 mg/L. Each bacterial suspension (50 μl) was also inoculated into a control well containing 50 μl Mueller Hinton broth (BD, Sparks, MD), with no antibiotic. The 96-well plates were incubated 18 hours at 37°C. The drug concentration in the first well with no visible growth was determined to be the MIC.

## DNA extraction and 16S rRNA gene sequencing

Fresh feces collected at each time point were frozen in 1 gram aliquots for future analysis. From these samples, 50 mg of feces were extracted individually using a MoBio PowerMag

Microbiome kit (Qiagen, Inc., Germanton, MD) optimized for the epMotion 5075 TMX (Eppendorf, Hauppauge, NY). The DNA libraries were prepared as described previously [24].

## Illumina MiSeq sequencing of bacterial communities

The V4 region of the 16S rRNA gene was amplified from each sample using the Dual indexing sequencing strategy [25]. Sequencing was done on the Illumina MiSeq platform, using a MiSeq Reagent Kit V2 500 cycles (2 x 250bp) (Illumina cat# MS102-2003), according to the manufacturer's instructions with modifications [25]. Accuprime High Fidelity Taq (Life Technologies cat # 12346094) was used. PCR was performed using the conditions (Standard or Touch Down) shown by Seekatz [24]. If additional template was used, the water volume was changed accordingly. PCR products were visualized using an E-Gel 96 with SYBR Safe DNA Gel Stain, 2% (Life technologies cat# G7208-02). Libraries were normalized using SequalPrep Normalization Plate Kit (Life technologies cat #A10510-01) following the manufacturer's protocol for sequential elution. The concentration of the pooled samples was determined using Kapa Biosystems Library Quantification kit for Illumina platforms (KapaBiosystems KK4824). The sizes of the amplicons in the library were determined using the Agilent Bioanalyzer High Sensitivity DNA analysis kit (cat# 5067–4626). The final library consisted of equal molar amounts from each of the plates, normalized to the pooled plate at the lowest concentration. Sequencing libraries were prepared according to Illumina's protocol for Preparing Libraries for Sequencing on the MiSeq (part# 15039740 Rev. D) for 2 nM or 4 nM libraries. If the library concentration was below 1 nM, an alternative method was used for denaturation [26]. PhiX and genomes were added in 16S amplicon sequencing to create diversity. Sequencing reagents were prepared according to the Schloss SOP [25], and custom read 1, read 2 and index primers were added to the reagent cartridge. FASTQ files were generated for paired end reads.

## Microbiota analysis

Analysis of the V4 region of the 16S rRNA gene was performed using the DADA2 method for the inference of exact amplicon sequence variants (ASVs) [27,28]. Our analysis protocol generally followed the DADA2 tutorial (https://benjjneb.github.io/dada2/tutorial.html). Samples were filtered based on a maximum expected error threshold of 2. ASVs were inferred using the dada function with default parameters except that samples were pooled in order to increase sensitivity to rare sequence variants. Chimeras were removed using the default consensus removal method [27]. Taxonomy was assigned using the implementation of the naive Bayesian classifier available in the dada2 R package, and the Silva v128 reference database [29,30].

## Statistical analysis

Mean pharmacokinetic parameters were compared using a Student's t test. Mean *E. coli* concentration was compared over time using one-way analysis of variance with the Holm-Sidak method for comparison of individual time points to time 0 [31,32] (SigmaPlot 14.0, Systat Software Inc., San Jose, CA). Descriptive statistics (mean, median, standard deviation, and range) for MIC were computed overall, by treatment and by time point. The dependent variable consisted of the MIC values of isolates (up to 8 isolates per sample and per time point) and independent variables included treatment (CHCL vs CCFA), time point (0, 24, 36, 48, 60, 72, 96, 120, 168, 240, and 336 hours) and a two-way interaction term between treatment and time point. Several family distributions and transformations of the outcome were modeled including normal, lognormal, tobit, poisson, and negative binomial. After checking assumptions, model fit was assessed using information criteria (AIC, BIC) and residual plots. The effect of treatment and time period on MICs (log10-based transformed) was estimated using generalized linear

mixed models (GLMMs) fitted with a Gaussian distribution, identity link, residual pseudo-like-lihood, Newton-Raphson with ridging optimization and Kenward Rogers adjustment for denominator degrees of freedom using Proc Glimmix in SAS (SAS 9.4, SAS Institute Inc., Cary, NC). A random intercept for animal and an unstructured covariance structure were included to account for the design structure of the study (isolates nested within samples (animals) and repeated measures at the animal level). Model assumptions were tested and residuals were investigated using graphical tools. Mean MIC values and their 95% confidence intervals were computed by drug and sample time. P-values < 0.05 were considered statistically significant. The Tukey-Kramer procedure was used to prevent inflation of Type I error due to multiple comparisons [33,34].

# Results

## Pharmacokinetic modeling

The values for pharmacokinetic parameters for both CHCL and CCFA are presented in Table 1. Because of the slow release of ceftiofur from CCFA, the maximum concentrations ($C_{MAX}$) of ceftiofur and metabolites in the plasma were significantly lower than from injection of CHCL (Fig 1; P = 0.01), while the $T_{MAX}$ (P<0.001), half-life (P<0.001), and AUC (P<0.001) were greater for steers receiving CCFA. Low drug concentrations in the GIT were noted over a longer time as the half-life of ceftiofur and metabolites were 2–3 times greater in the ileum and colon for CCFA than for CHCL, but due to the variability between animals this difference was not statistically significant. Similarly, the $T_{MAX}$ was later in both the ileum and colon of CCFA treated calves, although this difference was only significant in the colon (P = 0.03). The penetration of ceftiofur and metabolites into the ileum and colon were similar for both drugs at approximately 20% of plasma AUC. The penetration of CCFA into the ISF was significantly higher for CCFA (86 ± 62%) compared to CHCL (24 ± 16%; P = 0.009).

## Concentration of *E. coli*

The fecal concentration of *E. coli* was not different between CHCL (7.8 ± 0.25 $\log_{10}$ CFU/g) and CCFA (7.6 ± 0.35 $\log_{10}$ CFU/g) treatment groups at time 0 (P = 0.9). In the CHCL group, the mean concentration decreased by 1.7 $\log_{10}$ by 24 hours, but the change at this or any other time point was not significantly different compared to time 0 (P = 0.52). In the CCFA group, the mean concentration significantly decreased within 24 hours (5.4 + 0.38 $\log_{10}$ CFU/g, P = 0.002), and ultimately decreased by 3.4 $\log_{10}$ by 48 hours (P = 0.007). The concentration slowly increased after this point, but never returned to baseline (Fig 2).

## *E. coli* minimum inhibitory concentration

Descriptive statistics (mean, median, SD and range) for MIC values by drug and sample time are presented in Table 2. Table 3 depicts the mean MIC estimates from multivariable models including fixed effects for drug, sample time and drug by sample time. The interaction between drug formulation and sample time was significantly (P < 0.001) associated with MIC values, indicating that the effect of drug formulation on MIC values depended on the time point. When CCFA was administered, MIC values significantly increased at 24 hours (mean MIC = 1.32 ug/ml, 95% CI = 0.60–1.93 ug/ml) and continued to increase up to 96 hours, when MIC peaked (mean MIC = 10.68 ug/ml, 95% CI = 5.47–20.85 ug/ml), followed by a decrease in MIC values to baseline levels, and the $MIC_{50}$ was within the wild-type distribution at 14 days after treatment. The mean MIC was significantly greater than the MIC at 0 hours from 24 hours through 168 hours after treatment (Table 3). When CHCL was administered, MIC

**Table 1. Pharmacokinetic parameters for ceftiofur crystalline free acid (CCFA) and ceftiofur hydrochlroide (CHCL) in the plasma, interstitial fluid (ISF), ileum and colon of steers.**

**CCFA Plasma**

| Parameter | Units | Mean | Std Dev | CV% |
|---|---|---|---|---|
| AUC | hr*ug/ml | 182.26 | 22.97 | 12.60 |
| $C_{MAX}$ | ug/ml | 1.80 | 0.95 | 52.70 |
| k01 | 1/hr | 0.26 | 0.10 | 37.50 |
| Absorption T½ | hr | 3.07 | 1.16 | 37.81 |
| k10 | 1/hr | 0.01 | 0.01 | 49.66 |
| Elimination T½ | hr | 73.25 | 33.54 | 45.79 |
| $T_{MAX}$ | hr | 14.13 | 4.56 | 32.25 |

**CHCL Plasma**

| Parameter | Units | Mean | Std Dev | CV% |
|---|---|---|---|---|
| AUC | hr*ug/ml | 61.63* | 11.63 | 18.88 |
| $C_{MAX}$ | ug/mL | 3.29* | 0.74 | 22.43 |
| k01 | 1/hr | 0.53 | 0.45 | 84.82 |
| Absorption T½ | hr | 1.80* | 0.74 | 40.92 |
| k10 | 1/hr | 0.08* | 0.01 | 12.30 |
| Elimination T½ | hr | 8.79* | 1.12 | 12.79 |
| $T_{MAX}$ | hr | 4.98* | 1.51 | 30.23 |

**CCFA ISF**

| Parameter | Units | Mean | Std Dev | CV% |
|---|---|---|---|---|
| AUC infinity | hr*ug/ml | 91.54 | 35.98 | 39.31 |
| AUC 0 to Cn | hr*ug/ml | 90.81 | 28.50 | 31.39 |
| $C_{MAX}$ | ug/ml | 1.62 | 0.61 | 37.68 |
| Half-life | hr | NA | NA | NA |
| Lambda z | 1/hr | NA | NA | NA |
| MRT | hr | NA | NA | NA |
| $T_{MAX}$ | hr | 76.67 | 65.99 | 86.07 |
| Penetration | | 0.86 | 0.62 | 71.44 |

**CHCL ISF**

| Parameter | Units | Mean | Std Dev | CV% |
|---|---|---|---|---|
| AUC_TAU | hr*ug/ml | 12.26 | 11.61 | 94.66 |
| AUC infinity | hr*ug/ml | 73.32 | 84.09 | 114.68 |
| AUC 0 to Cn | hr*ug/ml | 43.36* | 35.24 | 81.28 |
| $C_{AVE}$ | ug/ml | 0.51 | 0.48 | 94.66 |
| $C_{MAX}$ | ug/ml | 0.72* | 0.61 | 83.88 |
| $C_{MIN}$ | ug/ml | 0.40 | 0.48 | 119.17 |
| Half-life | hr | NA | NA | NA |
| Lambda z | 1/hr | NA | NA | NA |
| MRT | hr | NA | NA | NA |
| $T_{MAX}$ | hr | 5.00 | 4.69 | 93.81 |
| Penetration | | 0.24* | 0.16 | 65.59 |

**CCFA Ileum**

| Parameter | Units | Mean | Std Dev | CV% |
|---|---|---|---|---|
| AUC infinity | hr*ug/ml | 55.19 | 6.86 | 12.43 |
| AUC 0 to Cn | hr*ug/ml | 27.67 | 7.36 | 26.59 |
| $C_{MAX}$ | ug/ml | 0.54 | 0.16 | 30.37 |
| Half-life | hr | 127.74 | 16.46 | 12.89 |
| Lambda z | 1/hr | 0.01 | 0.00 | 12.16 |
| MRT | hr | 192.91 | 36.54 | 18.94 |
| $T_{MAX}$ | hr | 45.33 | 61.12 | 134.82 |
| Penetration | | 0.20 | 0.07 | 37.79 |

**CHCL Ileum**

| Parameter | Units | Mean | Std Dev | CV% |
|---|---|---|---|---|
| AUC_TAU | hr*ug/ml | 13.82 | 9.27 | 67.08 |
| AUC infinity | hr*ug/ml | 62.98 | 21.22 | 33.70 |
| AUC 0 to Cn | hr*ug/ml | 39.05 | 19.70 | 50.43 |
| $C_{AVE}$ | ug/mL | 0.58 | 0.39 | 67.08 |
| $C_{MAX}$ | ug/mL | 1.20 | 1.00 | 82.75 |
| $C_{MIN}$ | ug/mL | 0.06 | 0.10 | 154.92 |
| Half-life | hr | 66.27 | 79.51 | 119.97 |
| Lambda z | 1/hr | 0.06 | 0.06 | 97.53 |
| MRT | hr | 204.42 | 175.78 | 85.99 |
| $T_{MAX}$ | hr | 8.33 | 4.80 | 57.63 |
| Penetration | | 0.23 | 0.10 | 44.87 |

**CCFA Colon**

| Parameter | Units | Mean | Std Dev | CV% |
|---|---|---|---|---|
| AUC infinity | hr*ug/ml | 53.34 | 43.33 | 81.22 |
| AUC 0 to Cn | hr*ug/ml | 32.38 | 28.26 | 87.27 |
| $C_{MAX}$ | ug/ml | 0.44 | 0.24 | 54.82 |
| Half-life | hr | 94.85 | 39.34 | 41.47 |
| Lambda z | 1/hr | 0.01 | 0.01 | 64.74 |
| MRT | hr | 144.43 | 55.28 | 38.27 |
| $T_{MAX}$ | hr | 25.60 | 28.37 | 110.82 |
| Penetration | | 0.24 | 0.27 | 111.16 |

**CHCL Colon**

| Parameter | Units | Mean | Std Dev | CV% |
|---|---|---|---|---|
| AUC $_{TAU}$ | hr*ug/ml | 12.07 | 8.09 | 67.00 |
| AUC infinity | hr*ug/ml | 41.09 | 25.25 | 61.46 |
| AUC 0 to Cn | hr*ug/ml | 27.21 | 11.25 | 41.36 |
| $C_{AVE}$ | ug/ml | 0.50 | 0.34 | 67.00 |
| $C_{MAX}$ | ug/ml | 1.55 | 2.07 | 133.04 |
| $C_{MIN}$ | ug/ml | 0.02 | 0.03 | 137.15 |
| Half-life | hr | 39.45 | 42.39 | 107.45 |
| Lambda z | 1/hr | 0.04 | 0.04 | 80.86 |
| MRT | hr | 98.83 | 78.43 | 79.37 |
| $T_{MAX}$ | hr | 8.00* | 2.45 | 30.62 |
| Penetration | | 0.15 | 0.07 | 47.90 |

k01, and k10, rates for absorption and elimination processes, respectively, and accompanying half-lives (T½); AUC, area under the curve; AUC infinity, area under the curve from time zero to infinity; AUC 0 to Cn, area under the curve from time zero to the last measured time point (Cn); $AUC_{TAU}$ , AUC (τ) for the dose interval (tau = 24 hours) for ceftiofur administered 3 times; $C_{MAX}$, maximum drug concentration; $T_{MAX}$, time to maximum drug concentration; $C_{MIN}$ , minimum drug concentration; $C_{AVE}$ , average drug concentration; Lambda-z ($λ_Z$), terminal slope; MRT, mean residence time; penetration factor, calculated from the AUC ratios of tissue fluid/plasma; NA indicates that there was insufficient sample collection to calculate these values;

* indicates that values are significantly different between the two formulations, P<0.05.

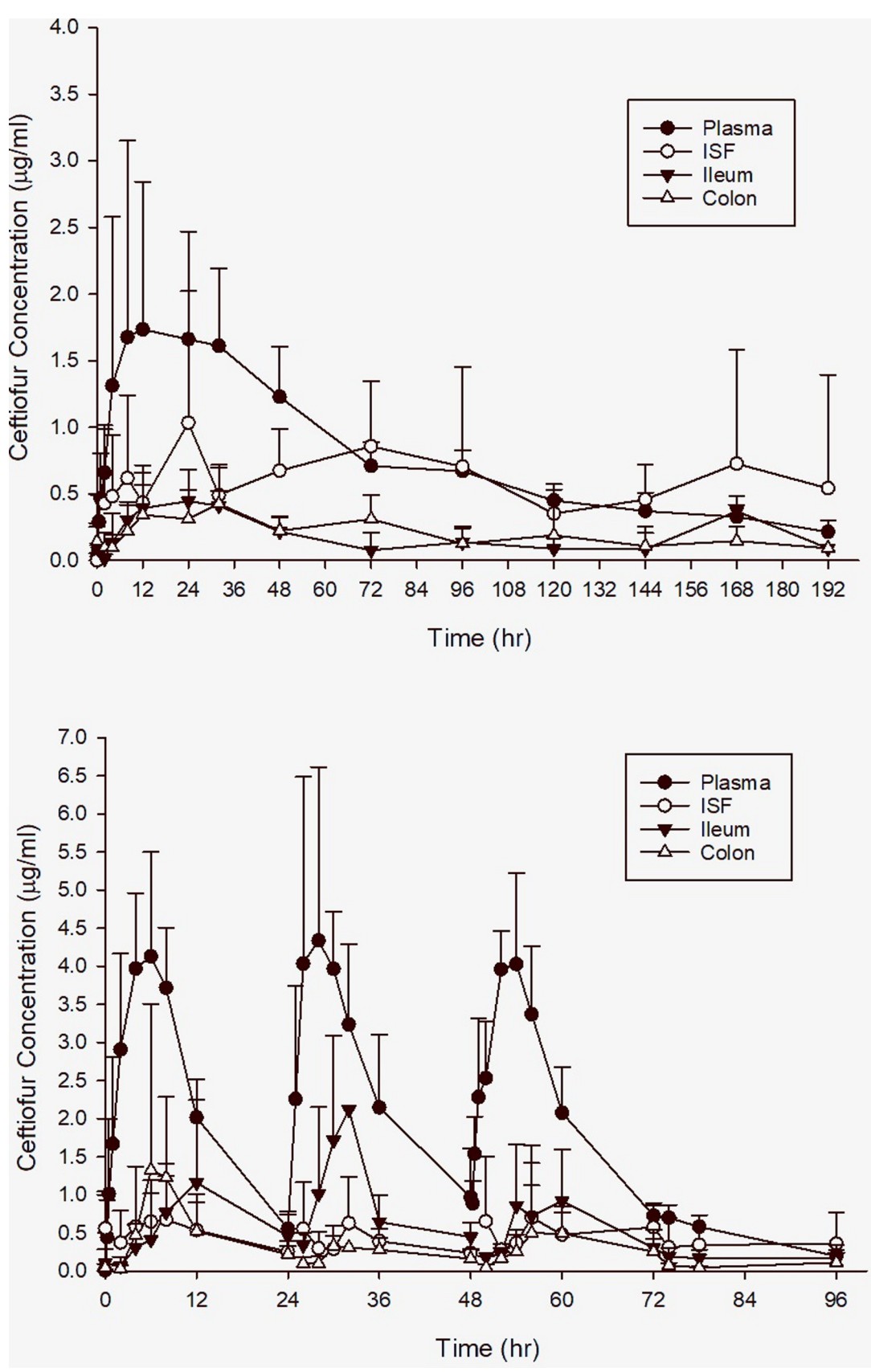

**Fig 1. Total concentration of ceftiofur equivalents in plasma, interstitial fluid (ISF), ileum, and colon for steers treated with (A) ceftiofur crystalline free acid (CCFA) and (B) ceftiofur hydrochloride (CHCL).**

values peaked at 48 hours (mean MIC = 20.45 ug/ml, 95% CI = 10.29–40.63 ug/ml) and then again at 72 hours (mean MIC = 19.58 ug/ml, 95% CI = 10.03–38.24 ug/ml), followed by a steady decrease to baseline levels, and the MIC$_{50}$ was within the wild-type distribution by 120 hours after the initial dose (Table 3 and Fig 3). In this group, the mean MIC was significantly greater than the MIC at 0 hours from 24 hours to 96 hours after initial treatment (Table 3). At no time was there a significant difference between the two treatment groups at the same time point.

## Alterations in the fecal microbiota

As seen in Fig 4, there was a shift in the microbial communities after treatment with either CHCL or CCFA. This shift appears to be slower, but more pronounced and persistent in the steers treated with CCFA. Yet, in neither group does the community return to its initial structure at 2 weeks after treatment. These shifts are largely due to a reduction in Firmicutes and an increase in Bacteroidetes (Fig 5). The Archea, primarily composed of *Methanobrevibacter*, follow a similar trajectory of initial decline with a slow, incomplete recovery over time (Fig 6).

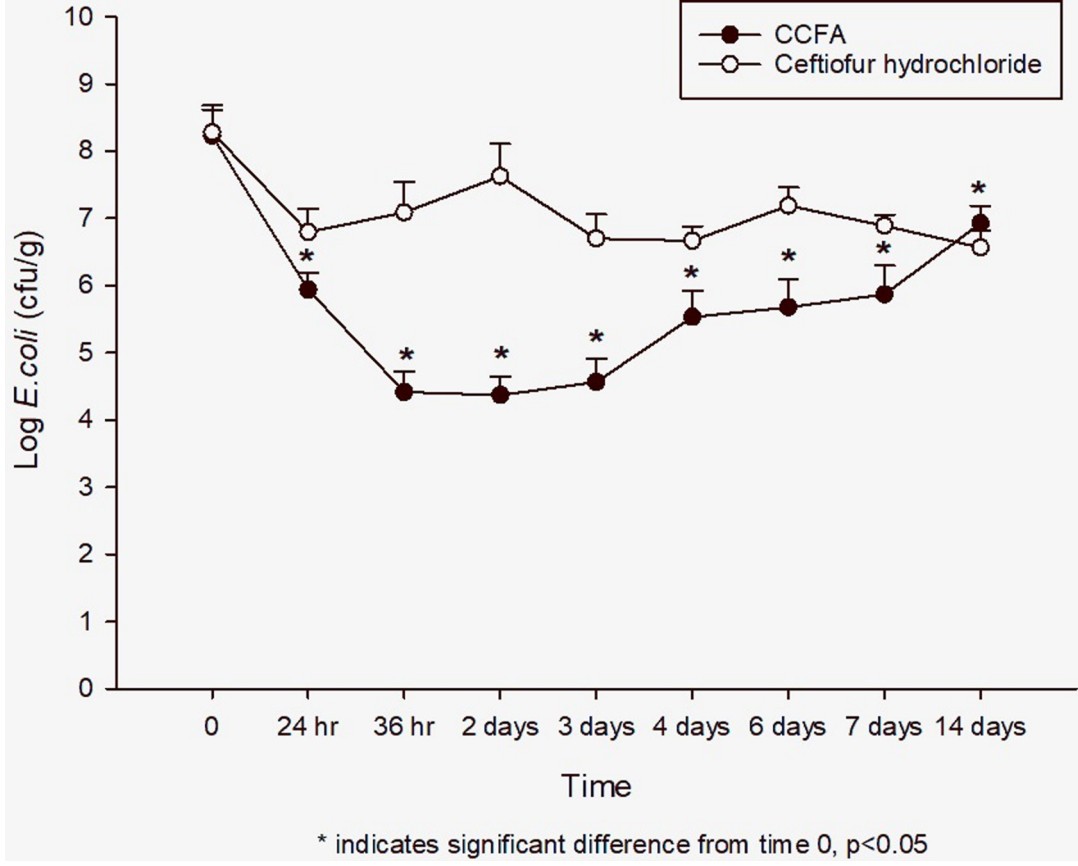

**Fig 2. Fecal *E. coli* concentration over time after treatment with either ceftiofur crystalline free acid (CCFA) or ceftiofur hydrochloride (CHCL).** Mean ± SD. * indicates a significant difference from time 0, P ≤0.05.

**Table 2. Descriptive statistics for minimum inhibitory concentration (MIC) by drug and time point.**

| Drug | Time Point | n | Mean | Median | SD | Range |
|------|-----------|---|------|--------|-----|-------|
| | | | **MIC** | | | |
| CCFA | | | | | | |
| | 0 h | 48 | 0.46 | 0.50 | 0.13 | 0.25–1.00 |
| | 24 h | 48 | 14.5 | 0.50 | 51.33 | 0.25–256.00 |
| | 36 h | 48 | 13.16 | 0.50 | 38.41 | 0.25–256.00 |
| | 48 h | 48 | 23.63 | 8.00 | 61.14 | 0.25–256.00 |
| | 60 h | - | - | - | - | - |
| | 72 h | 48 | 35.58 | 12.00 | 76.26 | 0.50–256.00 |
| | 96 h | 48 | 37.16 | 16.00 | 75.75 | 0.25–256.00 |
| | 120 h | 48 | 23.57 | 8.00 | 61.12 | 0.25–256.00 |
| | 168 h | 48 | 32.33 | 8.00 | 77.30 | 0.25–256.00 |
| | 240 h | - | - | - | - | - |
| | 336 h | 48 | 1.05 | 0.50 | 1.84 | 0.25–8.00 |
| CHCL | | | | | | |
| | 0 h | 48 | 8.75 | 0.50 | 37.03 | 0.25–256.00 |
| | 24 h | 48 | 8.71 | 8.00 | 9.20 | 0.25–32.00 |
| | 36 h | 48 | 12.31 | 16.00 | 8.53 | 0.25–32.00 |
| | 48 h | 48 | 41.85 | 16.00 | 74.09 | 1.00–256.00 |
| | 60 h | 48 | 32.52 | 16.00 | 59.70 | 0.25–256.00 |
| | 72 h | 48 | 50.85 | 16.00 | 85.99 | 0.25–256.00 |
| | 96 h | 48 | 51.05 | 8.00 | 93.11 | 0.25–256.00 |
| | 120 h | 48 | 5.31 | 0.50 | 6.70 | 0.25–16.00 |
| | 168 h | 48 | 10.47 | 0.50 | 37.10 | 0.25–256.00 |
| | 240 h | 48 | 7.89 | 0.50 | 36.91 | 0.25–256.00 |
| | 336 h | 48 | 2.17 | 0.50 | 4.61 | 0.25–16.00 |

CCFA = ceftiofur crystalline free acid; CHCL = ceftiofur hydrochloride; n = number of observations; SD = standard deviation.

## Discussion

Because of its broad spectrum of activity, short slaughter withdrawal time, and zero milk withdrawal time, ceftiofur is one of the most commonly used antimicrobials in cattle in the United States. As it is available in multiple formulations for use in cattle, we investigated the gastrointestinal PK of two different formulations and their impact on enteric bacteria to determine if selection of one formulation over another could be a viable means to mitigate selection of AMR enteric bacteria.

Because of the slow release of ceftiofur from CCFA, the maximum concentrations ($C_{MAX}$) of ceftiofur and metabolites in the plasma were significantly lower than from injection of CHCL (Fig 1). The slow release formulation also impacted the plasma $T_{MAX}$ as concentrations from CCFA peaked later than that of CHCL, but the prolonged half-life significantly increased the plasma AUC. These findings are similar to a comparative PK study in neonatal calves [35]. ISF fluid concentrations for CHCL were similar to those of ceftiofur sodium [15], and reflects the high protein binding (93%) of the metabolite measured in our previous study [22]. However, the ISF concentrations after CCFA injection were much higher. This likely occurred because of a longer time for equilibration between plasma and interstitial tissue fluid for CCFA. This also produced longer persistence of ceftiofur and its metabolites in intestinal fluids. These observations are consistent with previous evidence from tissue cages that showed

**Table 3. Model-adjusted mean\*minimum inhibitory concentration (MIC) estimates, 95% confidence intervals and P-values by drug, sample time and drug by sample time.**

| Variable | Mean MIC | 95% CI mean MIC | P-value[†] |
|---|---|---|---|
| Drug | | | 0.425 |
| CCFA | NA | NA | |
| CHCL | 3.53 | 1.91–6.52 | |
| Sample Time | | | <0.001 |
| 0 | 0.69 | 0.46–1.05 | |
| 24 | 1.93 | 1.19–3.14 | |
| 36 | 3.56 | 2.20–5.77 | |
| 48 | 8.98 | 5.53–14.58 | |
| 60 | - | - | |
| 72 | 12.88 | 7.97–20.84 | |
| 96 | 9.11 | 5.44–15.26 | |
| 120 | 2.40 | 1.39–4.12 | |
| 168 | 2.48 | 1.51–4.08 | |
| 240 | - | - | |
| 336 | 0.61 | 0.39–0.97 | |
| Drug x Sample Time | | | < 0.001 |
| CCFA 0 | 0.44 | 0.25–0.80 | |
| CCFA 24 | 1.32 | 0.60–1.93 | |
| CCFA 36 | 2.12 | 1.07–4.19 | |
| CCFA 48 | 3.94 | 1.98–7.83 | |
| CCFA 60 | - | - | |
| CCFA 72 | 8.48 | 4.25–16.91 | |
| CCFA 96 | 10.68 | 5.47–20.85 | |
| CCFA 120 | 3.67 | 1.67–8.04 | |
| CCFA 168 | 4.24 | 2.01–8.96 | |
| CCFA 240 | - | - | |
| CCFA 336 | 0.61 | 0.32–1.70 | |
| CHCL 0 | 1.08 | 0.60–1.93 | |
| CHCL 24 | 2.83 | 1.42–5.62 | |
| CHCL 36 | 5.99 | 3.03–11.84 | |
| CHCL 48 | 20.45 | 10.29–40.63 | |
| CHCL 60 | 14.89 | 7.46–29.70 | |
| CHCL 72 | 19.58 | 10.03–38.24 | |
| CHCL 96 | 7.77 | 3.55–20.85 | |
| CHCL 120 | 1.57 | 0.74–3.31 | |
| CHCL 168 | 1.46 | 0.76–2.79 | |
| CHCL 240 | 0.90 | 0.48–1.70 | |
| CHCL 336 | 0.61 | 0.32–1.18 | |
| Significant contrasts for Drug x Sample Time interaction | | | |
| Contrast | P-value[‡] | | |
| CCFA 0 vs CCFA 24 | 0.009 | | |
| CCFA 0 vs CCFA 36 | <0.001 | | |
| CCFA 0 vs CCFA 48 | <0.001 | | |
| CCFA 0 vs CCFA 72 | <0.001 | | |
| CCFA 0 vs CCFA 96 | <0.001 | | |
| CCFA 0 vs CCFA 120 | <0.001 | | |

(*Continued*)

**Table 3.** (Continued)

| Variable | Mean MIC | 95% CI mean MIC | P-value[†] |
|---|---|---|---|
| CCFA 0 vs CCFA 168 | <0.001 | | |
| CHCL 0 vs CHCL 24 | 0.041 | | |
| CHCL 0 vs CHCL 36 | <0.001 | | |
| CHCL 0 vs CHCL 48 | <0.001 | | |
| CHCL 0 vs CHCL 60 | <0.001 | | |
| CHCL 0 vs CHCL 72 | <0.001 | | |
| CHCL 0 vs CHCL 96 | <0.001 | | |

CCFA = ceftiofur crystalline free acid; CHCL = ceftiofur hydrochloride; NA = Not available.

† Overall significance test (F-test) .

‡ P-values represent Tukey-Kramer's adjustment for multiple comparisons.

*These estimates are from GLMM models including drug, sample time and a two-way. interaction between drug and sample time, after accounting for isolates nested within samples and repeated measures at the animal level.

that with a prolonged half-life, penetration into interstitial fluids can increase due to the additional time for diffusion [36,37].

The concentrations measured in the GIT were lower than previously reported [15], which may be due to differences in formulation, but we cannot confirm this without additional

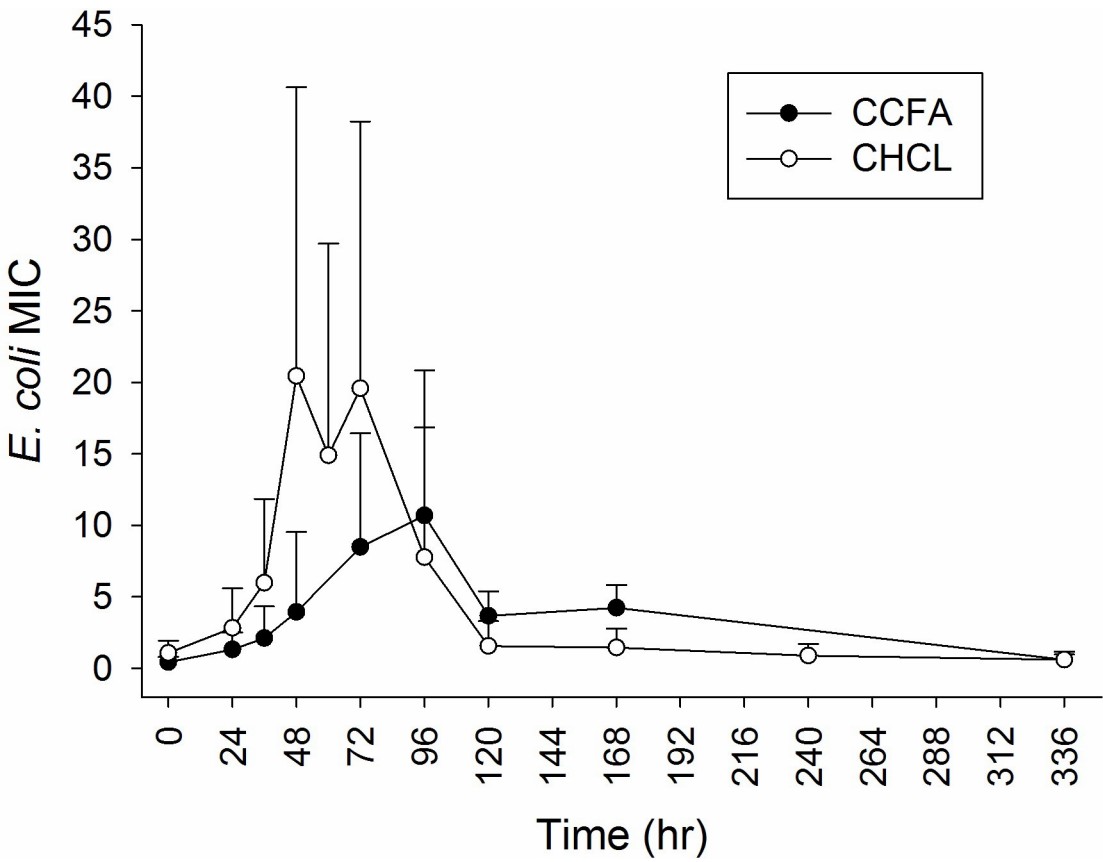

**Fig 3. Mean minimum inhibitory concentration (MIC, ± 95% CI) of ceftiofur in *E. coli* isolates from steers treated with ceftiofur crystalline free acid (CCFA) and ceftiofur hydrochloride (CHCL).**

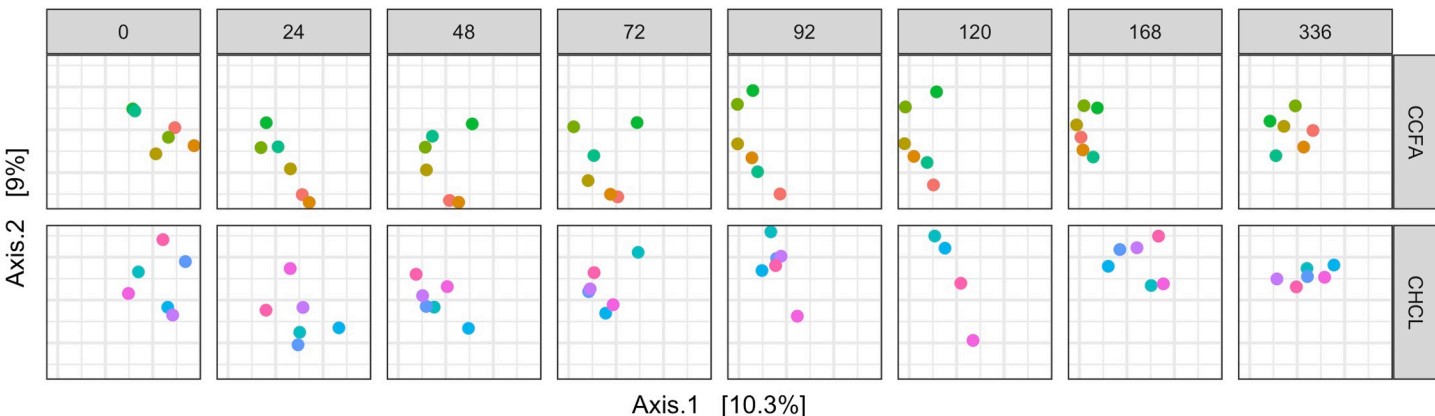

**Fig 4. Principal coordinate analysis of the microbial communities in each steer over time.**

study. In the previous study, steers were administered ceftiofur sodium, compared to CHCL and CCFA that were administered in this study. Interestingly, there minimal significant differences in the GI PK parameters between the two formulations in this study. This may be explained by the large variability between calves and the relatively small numbers in each group. The penetration into the ileum and colon was similar for both formulations, yet the numerical differences in $C_{MAX}$ and half-life suggest prolonged, low drug concentrations in the

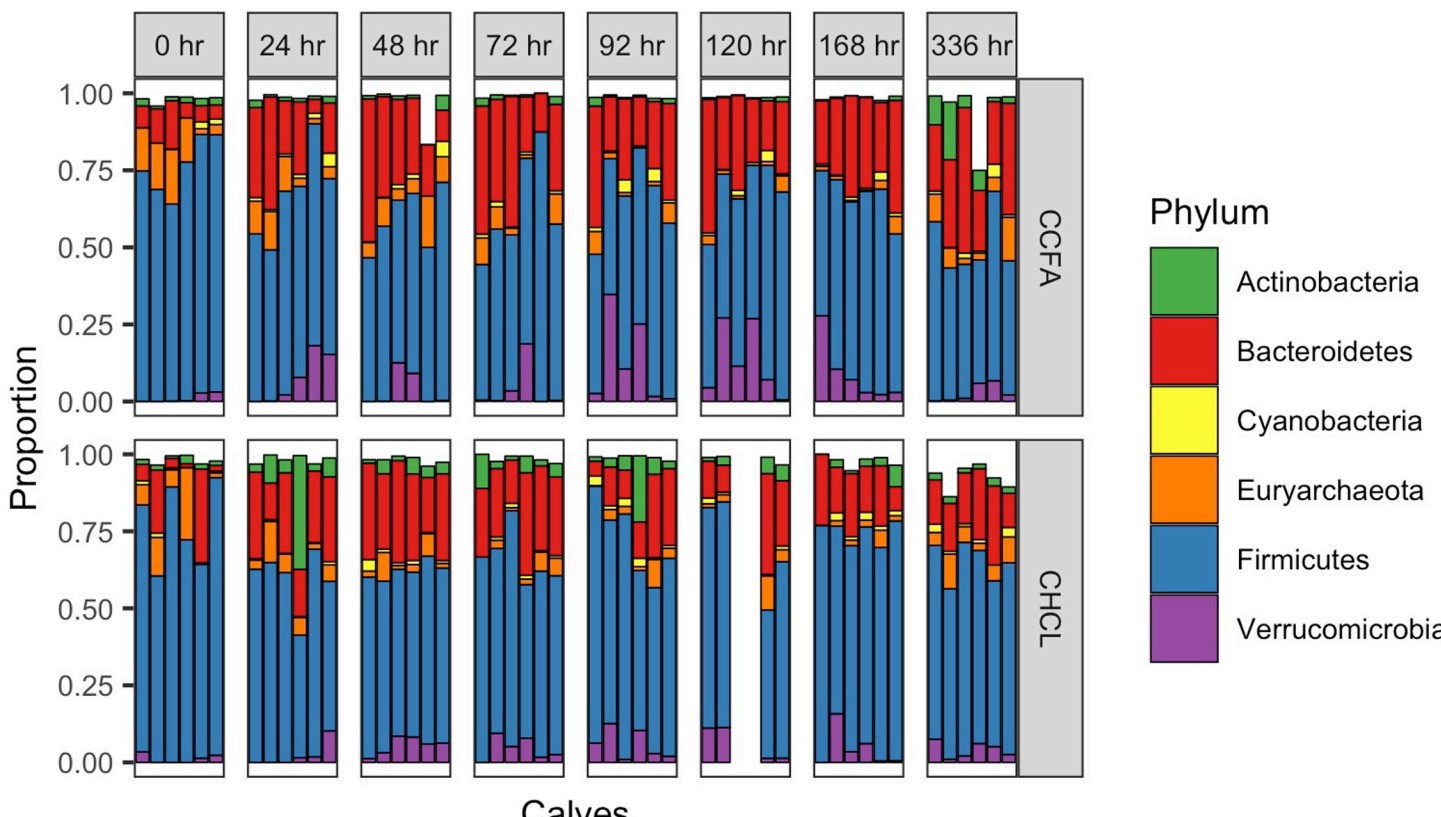

**Fig 5. The relative abundance of each phylum present in steers after treatment with either ceftiofur hydrochloride (CHCL) or ceftiofur crystalline free acid (CCFA).** Each bar represents an individual calf.

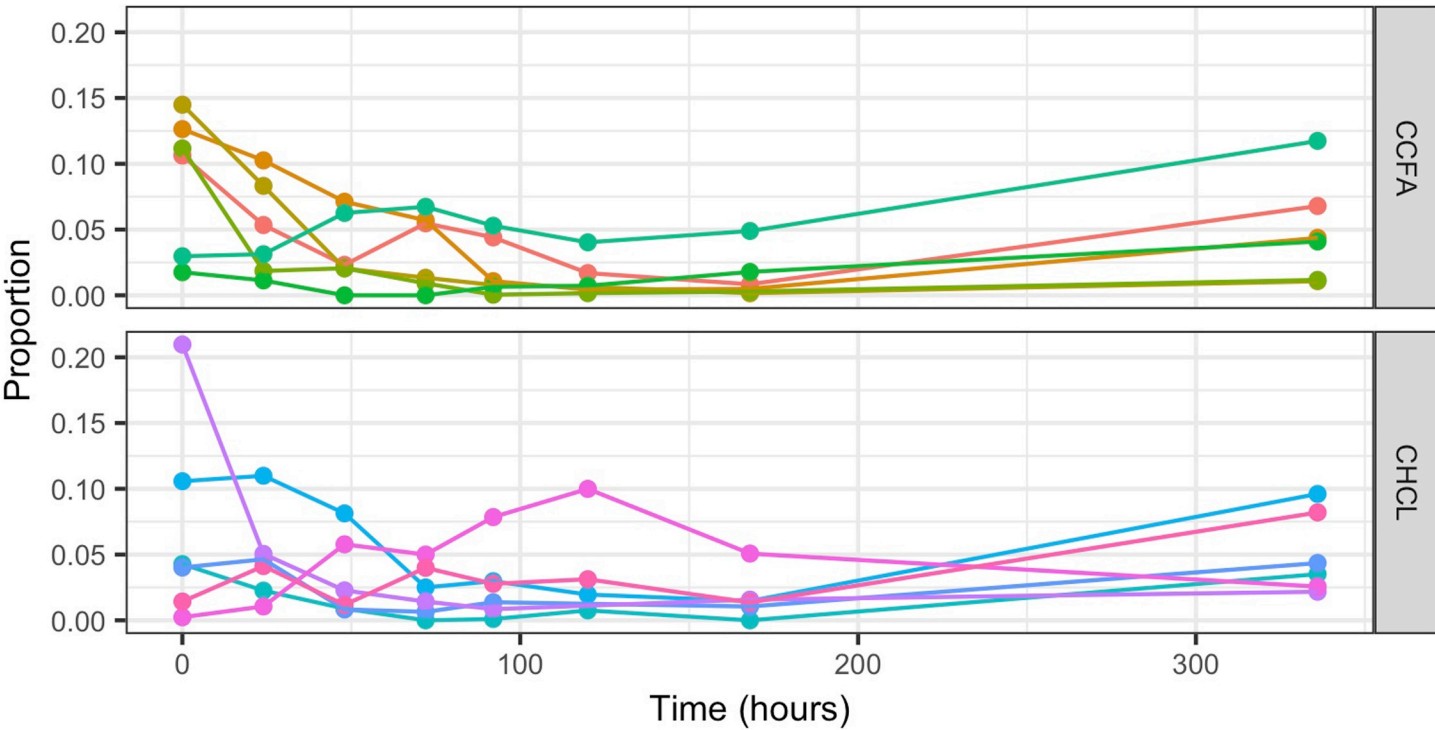

**Fig 6. Changes in the proportion of *Methanobrevibacter* over time in both the ceftiofur hydrochloride (CHCL) and ceftiofur crystalline free acid (CCFA) groups.**

ileum and colon of steers receiving CCFA. Because there were differences in the effect on *E. coli* and the microbiome, it suggests to us that this property produced clinically relevant differences in the GI PK parameters. Further, these concentrations were dramatically lower than predicted through mathematical models [38], demonstrating the need for empirical data. Here, the intestinal concentrations were above MIC 90 for *E. coli* after CHCL administration, but only briefly. For CCFA, concentrations never reached the MIC 90, but the concentrations were above the MIC for the most susceptible wild-type strains.

According to EUCAST (www.EUCAST.org) wild-type *E. coli* have ceftiofur MIC values that range from 0.12–1.0 µg/ml; thus, the more susceptible strains were exposed to ceftiofur and metabolites longer after injection of CCFA compared to CHCL. This may explain why CCFA had a more significant impact on the concentration of *E. coli* in the feces compared to CHCL. As ceftiofur is a time-dependent drug, one can speculate that longer drug exposure of *E. coli* in the GIT produced a greater reduction in the *E. coli* populations in spite of the low drug concentrations. This reduction is similar to what has been described previously in feedlot cattle [10,39]. CHCL peak concentrations in GIT were higher than CCFA, but with a much shorter half-life. As demonstrated in our previous study with ceftiofur sodium and by others, at this high concentration ceftiofur appears to be rapidly degraded by enteric bacteria, which shortens the exposure time [15,40]. This may mitigate the impact of the drug on the concentration of *E. coli* in the feces in our study. In a previous study of dairy cattle treated daily for 5 days with ceftiofur, there was a significant decrease in fecal shedding of *E. coli* by these cows [41]. It is unclear in that study if the cows received CHCL or ceftiofur sodium. We found higher concentrations of ceftiofur and metabolites in the intestine after injections of ceftiofur sodium [15] compared to CHCL in this study, which could explain this difference.

The relative impact of the two formulations on microbiota of the steers was similar to their impact on *E. coli*. CCFA appears to have a more significant and prolonged effect on the bacterial communities overall. Specifically, there is a greater reduction in Firmicutes and increase in Bacteroidetes in steers treated with CCFA compared to CHCL. The clinical impact of these changes is undetermined, because the normal microbiota is undefined in this population. Both phyla are commonly found in the feces of adult cattle, with Firmicutes commonly being the predominate phylum [6,42]. Most studies of *Methanobrevibacter* have shown that it is the most common methanogen in the rumen of cattle [43,44], but its role in the fecal microbiota is unclear. Interestingly, reducing this organism could reduce methane production in treated animals and improve feed efficiency [43]. It is not known if this reduction in *Methanobrevibacter* is found in the rumen as well. Antimicrobial concentrations in the rumen after injection of these formulations have not been reported.

Though CCFA had a greater impact on the concentration of *E. coli* and the microbiome, the changes in *E. coli* ceftiofur MIC depended on the time of sampling, with no evidence of statistical differences in MIC between treatments at the same time point. Over time, the increase in mean MIC was greater in the CHCL group than in CCFA. The increase in mean MIC persisted longer for CCFA than CHCL (168 hours vs 96 hours), which is not surprising as the drug in intestinal fluids persisted longer from CCFA than CHCL (Fig 1). Nonetheless, in both groups, the MIC values returned to baseline prior to the end of meat residue withdrawal time for each drug, suggesting that persistence of *E. coli* with an MIC above the wild-type cutoff (www.EUCAST.org) in an animal at slaughter would be relatively unlikely. These findings are similar to previous results in dairy cattle [13,41] and beef cattle [39] demonstrating short-lived resistance to third-generation cephalosporins. Resistance to third-generation cephalosporin among *E. coli* isolates found in cattle at slaughter [45] is likely caused by other factors, rather than single uses of ceftiofur in cattle. Two studies in feedlot cattle have demonstrated a significant increase in resistant fecal *E. coli* [10] and carriage of cephalosporin resistance genes [11] in association with combined treatment with CCFA and oral chlortetracycline. This suggests that co-selection of resistance mechanisms may play a greater role in maintaining these resistance elements within the fecal microbiota once the initial selection pressure associated with ceftiofur administration has waned.

While this is the first study to associate intestinal pharmacokinetics with changes in AMR in enteric *E. coli* and with changes in the microbiome, our conclusions are limited by the size of the study. The sample size was determined based on the numbers needed to assess the pharmacokinetics of the drugs and this may not have been adequate for the microbial analyses. When analyzing MIC values, challenges in terms of the nature of the distribution of these data, which are not truly continuous and may be truncated, arise. Although several family distributions and transformations of the outcome were attempted, the chosen logarithmic transformation and statistical model provided a better fit by improving the skewness of the underlying distribution while accounting for the design structure (lack of independence due to multiple isolates per sample and repeated measures) of the study. Statistical comparisons of the changes in the microbiome are limited due to the high variability and small sample size. Our observations on antimicrobial resistance are limited to the changes in *E. coli*. Although this organism is commonly used as an indicator organism, it is unclear how generalizable these findings are to changes in the susceptibility profile of other enteric bacteria.

In conclusion, the relatively long persistence of active drug in the intestine of cattle treated with CCFA has a significant and prolonged effect on the concentration of *E. coli* in the feces and the microbiome. Repeated injections of CHCL did not have the same effect on fecal *E. coli* concentrations or the microbiome. CCFA increased the mean MIC of ceftiofur in fecal *E. coli* for a longer period of time, but this returned to baseline with two weeks after treatment.

## Acknowledgments

This research was supported by work performed by The University of Michigan Microbial Systems Molecular Biology Laboratory. The authors thank Delta R. Dise of the NC State University Clinical Pharmacology Laboratory for her expertise performing the drug assays.

## Author Contributions

**Conceptualization:** Derek M. Foster, Megan E. Jacob, Casey M. Theriot, Timo Prange, Mark G. Papich.

**Data curation:** Megan E. Jacob, Kyle A. Farmer, Benjamin J. Callahan, Casey M. Theriot, Natalia Cernicchiaro, Mark G. Papich.

**Formal analysis:** Derek M. Foster, Benjamin J. Callahan, Casey M. Theriot, Natalia Cernicchiaro, Mark G. Papich.

**Funding acquisition:** Derek M. Foster, Megan E. Jacob, Casey M. Theriot, Sophia Kathariou, Mark G. Papich.

**Investigation:** Derek M. Foster, Megan E. Jacob, Kyle A. Farmer, Benjamin J. Callahan, Casey M. Theriot, Sophia Kathariou, Mark G. Papich.

**Methodology:** Derek M. Foster, Megan E. Jacob, Benjamin J. Callahan, Casey M. Theriot, Timo Prange, Mark G. Papich.

**Project administration:** Derek M. Foster, Megan E. Jacob, Timo Prange, Mark G. Papich.

**Resources:** Derek M. Foster.

**Supervision:** Derek M. Foster, Megan E. Jacob.

**Validation:** Natalia Cernicchiaro.

**Visualization:** Benjamin J. Callahan, Natalia Cernicchiaro.

**Writing – original draft:** Derek M. Foster, Benjamin J. Callahan, Natalia Cernicchiaro, Mark G. Papich.

**Writing – review & editing:** Derek M. Foster, Megan E. Jacob, Benjamin J. Callahan, Casey M. Theriot, Sophia Kathariou, Natalia Cernicchiaro, Timo Prange, Mark G. Papich.

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
