## [Decision Letter · Decision Letter 0]

1 Jul 2019

PONE-D-19-14785

Ceftiofur formulation differentially affects the intestinal drug concentration, resistance of fecal Escherichia coli, and the microbiome of steers

PLOS ONE

Dear Dr. Foster,

Thank you for submitting your manuscript to PLOS ONE. After careful consideration, we feel that it has merit but does not fully meet PLOS ONE’s publication criteria as it currently stands. The manuscript has been reviewed by two experts in the field and they have raised concerns because of the experimental design, statistical analyses and the conclusions drawn from the results. They also indicated discrepancies between text and tables, while the contents of the tables should be precise. Therefore, we invite you to submit a revised version of the manuscript that addresses the points raised during the review process.

We would appreciate receiving your revised manuscript by Aug 15 2019 11:59PM. To enhance the reproducibility of your results, we recommend that if applicable you deposit your laboratory protocols in protocols.io, where a protocol can be assigned its own identifier (DOI) such that it can be cited independently in the future. For instructions see: http://journals.plos.org/plosone/s/submission-guidelines#loc-laboratory-protocols

We look forward to receiving your revised manuscript.

Kind regards,

Kristin Mühldorfer

Academic Editor

PLOS ONE

**Journal Requirements:**

2. In your Methods, please state the volume of the blood samples collected for use in your study.

3. In your Methods section, please include a comment about the state of the animals following this research. Were they euthanized or housed for use in further research? If any animals were sacrificed by the authors, please include the method of euthanasia and describe any efforts that were undertaken to reduce animal suffering.

"I have read the journal's policy and the authors of this manuscript have the following competing interests: DMF has received research support from Zoetis. MGP has received gifts, honoraria, consulting fees, and research support from Zoetis, the manufacturer of ceftiofur.  "

**Comments to the Author**

1. Is the manuscript technically sound, and do the data support the conclusions?

Reviewer #1: Partly

Reviewer #2: Partly

2. Has the statistical analysis been performed appropriately and rigorously? 

Reviewer #1: I Don't Know

Reviewer #2: Yes

3. Have the authors made all data underlying the findings in their manuscript fully available?

Reviewer #1: Yes

Reviewer #2: Yes

4. Is the manuscript presented in an intelligible fashion and written in standard English?

Reviewer #1: Yes

Reviewer #2: Yes

5. Review Comments to the Author

Reviewer #1: Congratulations to the authors for this well planned animal study, with linkable effects of administration of ceftiofur in two formulations (Excede and Excenel) on amount and MIC of E. coli and impact on other microbiota in colon, combined with PK data at different sites (serum, ISF and Intestinal fluid) at identical time-points.

Some findings are contradictory with each other or with other, older studies.

The ISF concentrations after CCFA injection were much higher. This likely occurred because of a longer time for equilibration between plasma and interstitial tissue fluid for CCFA. But why is the half-live in ISF for CCFA much shorter than for CHCL? This is also contradictory to the next statement, that this longer time for equilibration is responsible for a longer persistence (implying and also showing longer half-lives) in the intestinal fluids. Why isn’t this persistence of CCFA found in ISF?

With the repetitive injections during three days of CHCL, with three times relatively high concentrations, one could expect to find E. coli with higher MIC’s. But this finding isn’t in line with the findings of Goessens et.al, Journal of Antimicrobial Chemotherapy (2007) 59, 507– 516. They showed that selection of resistant Enterobacter cloacae was induced by a regimen with more continuous administration of ceftazidime (every 6 hrs, to be compared with the continuous release of CCFA) as compared to 4 times higher dose every 24 hrs (comparable with the CHCL regimen). This is attributable to the period in the mutant selection window. Goessens et al did determine the mutant preventive concentration (MPC), which was 16 mg/L for E. cloacae. Looking at the concentrations attained with the (authorized) dosage regimens of both ceftiofur products, and assuming that the MPC for E. coli won’t be lower than 16 mg/L, one must conclude that both regimens (3 days 2.2 mg/kg and once 6.6 mg/kg) are prone to select during therapy, and therefor no preferred regimen can be determined. Knowing the MIC and the MPC (of ceftiofur) for E. coli, and adjusting the dosing regimen to these MIC and MPC, combined with the finding that the concentrations found in ISF and intestinal fluids for ceftiofur are not very different from the levels in plasma, could change the outcomes in selection of bacteria with high MICs.

L 43 ...is one of the most common antimicrobials...: please be more specific: might be applicable for the USA (but not in all countries worldwide)

L 239 Schloss SOP: please add a reference; my question: are all specifications mentioned here divergent form the SOP?

L 247 ...removed using the default consensus removal method....: Add ref and specify

L 253 ...Holm-Sidak method for comparison of individual time points to time 0..:. Add ref

L 270 ...The Tukey-Kramer procedure...: Add ref

L 285/table1:

specify AUC in plasma

CCFA ISF: AUCinf < AUV 0 to cn seems impossible? Half-live=17.81 hr vs 44.94 hr of CHCL (but Lambda z is same magnitude 0.5 and 0.4?), and still AUC CCFA > AUC CHCL?

CCFA ileum: MRT is smaller than CHCL ileum?

Legend table 1 is incomplete: MRT

L 365 ...ceftiofur is one of the most commonly used antimicrobials in cattle...: add in USA

Reviewer #2: Dear Authors,

This manuscript investigates “Ceftiofur formulation differentially affects the intestinal drug concentration, resistance of fecal Escherichia coli, and the microbiome of steers”. The topic of the study is in line with those dealt by the journal and also the organization of manuscript is in line with instructions of PLOS ONE. It is thought that the data obtained from this study, will contribute to veterinary practice. With this approach, this study is considered as an original and valuable research. I find the authors have made a good study design and I would like to read this paper on PLOS ONE, but have multiple suggestions to consider. It is determined that the part of this article about MIK is written in a very descriptive and orderly manner. However, there are deficiencies in the information written about pharmacokinetics. You need to be more specific about pharmacokinetic parameters as these could affect your outcome.

Introduction

Line 43-44: Although Ceftiofur has been approved for the treatment of respiratory diseases, you should not forget that it is also used in the treatment of Ecoli-associated diarrhea in calves. I think you should be more informative instead of just saying that the drugs are administered for treatment of respiratory disease.

Materials and methods

Line 88-91 - Your pharmacokinetic design is not explained. The naming of your study design implies that all the calves were enrolled simultaneously. Please be more descriptive to allow the reader to understand how your study design. Cross-over or parallel pharmacokinetic design???

Line 98- Blood samples were collected in which tubes (EDTA or Li-heparin)? Provide tube information.

Line 99-101- You need to be more specific about blood, gastrointestinal and intestinal fluid sample point to make the meaning clear to the readers. For example; at 0 (pre-treatment), 5, 10, 15, 20, 25, 30, and 45 min and 36 h post-dosing. Which sampling times you have used for pharmacokinetic?

Line 113-114- "The calves received 2 mg/kg of flunixin meglumine intravenously prior to surgery and 24 hours after surgery". Could this drug have caused a change in the pharmacokinetic profile of ceftiofur? What impact this might have had on data interpretation, if any?

Line 115 and 127- You need to be more specific about predetermined time points.

Line 150- Are you used the WinNonlin program for statistical difference. I don’t understand..

Line 251- Did you compare the pharmacokinetic parameters statistically?

Results

Line 274- Pharmacokinetic modelling- Did you write your results here without any statistical comparison?

Discussion: Discussion of the study is not sufficient: the pharmacokinetic parameters were ignored. After the statistical analysis, I suggest re-evaluation of the relevant parts of the pharmacokinetic parameters.

Line 364-367- This line is like a repetition of the sentences in Line 43-44. I suggest you to combine these in introduction. Since your hypothesis is especially about the efficacy of ceftiofur in intestinal flora, you can instead add articles about the use of ceftiofur in calves with diarrhea. For this reason, I suggest below articles could be added to the discussion section of this study.

1. Constable P. D. 2009. Treatment of calf diarrhea: antimicrobial and ancillary treatments. Vet. Clin. North Am. Food Anim. Pract. 25: 101–120.

2. Feray ALTAN, Kamil UNEY, Ayse ER, Gul CETIN, Burak DIK, Enver YAZAR, and Muammer ELMAS. Pharmacokinetics of ceftiofur in healthy and lipopolysaccharide-induced endotoxemic newborn calves treated with single and combined therapy. J Vet Med Sci. 2017 Jul; 79(7): 1245–1252.

Line 367-368- This information is new and should be moved to the introduction section in line 54.

Line 370-372- This line is like a repetition of the sentences in Line 54-60. I suggest you to delete in discussion.

Line 372-374- I think it would be more appropriate for you to discuss this sentence in a different way rather than repeating what you have written in introduction section.

Line 375-377- You can't tell these without any statistical analysis. Also the table you have presented is so confused that I cannot reach this conclusion by looking at your table.

Line 376- Cmax of CCFA in ISF was not lower than CCFA. Please be carefully. The data you type in the table should be consistent with what you type in the text.

Line 379-384- You were stated “The ISF concentrations after CCFA injection were much higher. This likely occurred because of a longer time for equilibration between plasma and interstitial tissue fluid for CCFA. This also produced longer persistence of ceftiofur and its metabolites in intestinal fluids.” You have indicated in your table that the half-life of CCFA is shorter than CHLC in ISF. Can you explain why the half-life and the concentration of CHLC is long and low in ISF.

Line 385-409- Ceftiofur are beta-lactam antimicrobial drugs. Activity of beta-lactams is time-dependent kill characteristics, and the most useful and predictable parameter for optimal bactericidal activity is %T>MIC. Use this parameter to give information about the susceptible bacteria for ceftiofur. You can’t this say it through MIC 90 and concentration. You should determine to %T>MIC for susceptible bacteria.I suggest below article could be added

Turnidge, J. D. (1998). The pharmacodynamics of beta‐lactams. Clinical Infectious Diseases, 27, 10–12.

Table 1. I think you need more explanation in this table. Why did you not compare the differences in plasma concentrations?

Figure 5. Calves?????

6. PLOS authors have the option to publish the peer review history of their article (what does this mean?). If published, this will include your full peer review and any attached files.

Reviewer #1: Yes: Ingeborg M. van Geijlswijk

Reviewer #2: No

---

## [Author Response · Author response to Decision Letter 0]

19 Aug 2019

Thank you to the editors and reviewers for their helpful comments. We have attempted to address all of the stated concerns or explain our rationale for where we disagree. The line numbers listed here in our responses correspond to the version of the manuscript without tracked changes. 

Journal Requirements:

AU: Our apologies for these errors. We have reviewed the guidelines and corrected the manuscript throughout.

2. In your Methods, please state the volume of the blood samples collected for use in your study.

AU: This has been added to Line 106.

3. In your Methods section, please include a comment about the state of the animals following this research. Were they euthanized or housed for use in further research? If any animals were sacrificed by the authors, please include the method of euthanasia and describe any efforts that were undertaken to reduce animal suffering.

AU: The following statement has been added to lines 102-103. “At the conclusion of the study and observation of the appropriate meat withdrawal time, all ultrafiltration probes and catheters were removed, and the steers were sold.

"I have read the journal's policy and the authors of this manuscript have the following competing interests: DMF has received research support from Zoetis. MGP has received gifts, honoraria, consulting fees, and research support from Zoetis, the manufacturer of ceftiofur. "

AU: Any competing interests do not affect our adherence to PLOS ONE policies on data sharing. This information is now included in our new cover letter. 

AU: The data has now been uploaded to NCBI Bioproject. The ID is PRJNA560079. 

Comments to the Author

Reviewer #1: Congratulations to the authors for this well planned animal study, with linkable effects of administration of ceftiofur in two formulations (Excede and Excenel) on amount and MIC of E. coli and impact on other microbiota in colon, combined with PK data at different sites (serum, ISF and Intestinal fluid) at identical time-points.

Some findings are contradictory with each other or with other, older studies.

The ISF concentrations after CCFA injection were much higher. This likely occurred because of a longer time for equilibration between plasma and interstitial tissue fluid for CCFA. But why is the half-live in ISF for CCFA much shorter than for CHCL? This is also contradictory to the next statement, that this longer time for equilibration is responsible for a longer persistence (implying and also showing longer half-lives) in the intestinal fluids. Why isn’t this persistence of CCFA found in ISF?

AU: We appreciate the reviewer’s comments and observations. We agree that ISF concentrations were probably higher from CCFA compared to the other formulation because of longer time for equilibrium for the slow-release product. The reviewer also pointed out discrepancies in the half-life between formulations in the ISF fluid. This occurred as an artifact of the analysis and was overlooked earlier. Occasionally the interstitial probes did not collect enough fluid, or became plugged or kinked and failed. Therefore, we sometimes had only sparse points for which to calculate the half-life. For the measurement of a mean half-life value for ISF after administration of ceftiofur hydrochloride, upon further review of our data, we note that the mean was heavily skewed by two animals that had very long half-lives and the coefficient of variation for this parameter was very large. In retrospect, we should not have included the half-life from these individuals. Therefore, we have modified our data to eliminate these animals from the analysis. See our revised tables. We thank the reviewer for this careful observation. 

With the repetitive injections during three days of CHCL, with three times relatively high concentrations, one could expect to find E. coli with higher MIC’s. But this finding isn’t in line with the findings of Goessens et.al, Journal of Antimicrobial Chemotherapy (2007) 59, 507– 516. They showed that selection of resistant Enterobacter cloacae was induced by a regimen with more continuous administration of ceftazidime (every 6 hrs, to be compared with the continuous release of CCFA) as compared to 4 times higher dose every 24 hrs (comparable with the CHCL regimen). This is attributable to the period in the mutant selection window. Goessens et al did determine the mutant preventive concentration (MPC), which was 16 mg/L for E. cloacae. Looking at the concentrations attained with the (authorized) dosage regimens of both ceftiofur products, and assuming that the MPC for E. coli won’t be lower than 16 mg/L, one must conclude that both regimens (3 days 2.2 mg/kg and once 6.6 mg/kg) are prone to select during therapy, and therefor no preferred regimen can be determined. Knowing the MIC and the MPC (of ceftiofur) for E. coli, and adjusting the dosing regimen to these MIC and MPC, combined with the finding that the concentrations found in ISF and intestinal fluids for ceftiofur are not very different from the levels in plasma, could change the outcomes in selection of bacteria with high MICs.

AU: While the study by Goessens et al. is a robust evaluation of the different dosing regimens of ceftazidime, drawing comparisons from that study to ours is difficult. Differences in PK between cattle and rodents are likely significant, and their associations between plasma drug concentrations and selection for resistance may be very different than associations between drug concentrations in the GI tract and resistance. It is unclear from their study how closely the GI concentrations mirrored the plasma concentrations, and it is probably not appropriate to assume that ceftazidime concentrations in rodents is similar to our data with ceftiofur in cattle. Further, dosing every 6 hours may achieve a relatively constant drug concentration, but we would expect that this would be higher than that achieved by a slow release formulation like CCFA. 

One of the explanations for the findings offered in the comments is the time that concentrations were in the “Mutant Selection Window”. However, we do not believe that a mutant selection window (or mutant selection concentration) exists for cephalosporins against Enterobacteriaceae. Resistance to the cephalosporins is usually conferred by production of beta-lactamase. We believe that bacteria acquire the ability to produce beta-lactamases via transfer of genetic elements, not through spontaneous mutations in treated animals. Therefore, we do not believe a mutant prevention concentration (or window) can be defined for ceftiofur against E. coli. Our view on this is shared by others (eg, Smith et al. Stretching the mutant prevention concentration (MPC) beyond its limits. Journal of Antimicrobial Chemotherapy. 2003 Jun 1;51(6):1323-5.).

L 43 ...is one of the most common antimicrobials...: please be more specific: might be applicable for the USA (but not in all countries worldwide)

AU: “in the United States” has been added to line 48.

L 239 Schloss SOP: please add a reference; my question: are all specifications mentioned here divergent form the SOP?

AU: Reference 25 in line 252 is added to clarify that this SOP is contained in the supplemental data of this reference. The specifications in this section are included in this SOP except where noted with references 24 and 26. 

L 247 ...removed using the default consensus removal method....: Add ref and specify

AU: This has now been appropriately referenced. 

L 253 ...Holm-Sidak method for comparison of individual time points to time 0..:. Add ref

AU: References have been added.

L 270 ...The Tukey-Kramer procedure...: Add ref

AU: References have been added.

L 285/table1:

specify AUC in plasma

AU: This has been added to line 297 to clarify that the penetration is based on plasma AUC, and is a term that we defined as the ratio of the AUC of the tissue (ISF, intestinal fluids) to the AUC of plasma.

CCFA ISF: AUCinf < AUV 0 to cn seems impossible? Half-live=17.81 hr vs 44.94 hr of CHCL (but Lambda z is same magnitude 0.5 and 0.4?), and still AUC CCFA > AUC CHCL? CCFA ileum: MRT is smaller than CHCL ileum?

AU: We thank the reviewer for picking this up. The reviewer is correct. This is not possible. The half-life estimates for the ISF was skewed by a few outliers and we have modified the tables to eliminate this value. The reviewer is correct that AUC 0 to Cn should not be greater than AUC 0 to infinity. This was due to an error in which some animals were included in the mean calculation, even though the samples were incomplete and should not have been included. We have revised our tables to fix these errors. We apologize for not reviewing this more closely prior to manuscript submission. 

Legend table 1 is incomplete: MRT

AU: Thank you for catching this oversight. It has been added to line 309. 

L 365 ...ceftiofur is one of the most commonly used antimicrobials in cattle...: add in USA

AU: This has been added to line 378-379.

Reviewer #2: Dear Authors,

This manuscript investigates “Ceftiofur formulation differentially affects the intestinal drug concentration, resistance of fecal Escherichia coli, and the microbiome of steers”. The topic of the study is in line with those dealt by the journal and also the organization of manuscript is in line with instructions of PLOS ONE. It is thought that the data obtained from this study, will contribute to veterinary practice. With this approach, this study is considered as an original and valuable research. I find the authors have made a good study design and I would like to read this paper on PLOS ONE, but have multiple suggestions to consider. It is determined that the part of this article about MIK is written in a very descriptive and orderly manner. However, there are deficiencies in the information written about pharmacokinetics. You need to be more specific about pharmacokinetic parameters as these could affect your outcome.

Introduction

Line 43-44: Although Ceftiofur has been approved for the treatment of respiratory diseases, you should not forget that it is also used in the treatment of Ecoli-associated diarrhea in calves. I think you should be more informative instead of just saying that the drugs are administered for treatment of respiratory disease.

AU: Additional information in lines 48-50 has been added to clarify that ceftiofur is also used to treat enteric disease.

Materials and methods

Line 88-91 - Your pharmacokinetic design is not explained. The naming of your study design implies that all the calves were enrolled simultaneously. Please be more descriptive to allow the reader to understand how your study design. Cross-over or parallel pharmacokinetic design???

AU: Lines 97-98 have been edited to clarify that it was parallel study was used. Use of a crossover design would have interfered with the interpretation of the microbiological results.

Line 98- Blood samples were collected in which tubes (EDTA or Li-heparin)? Provide tube information.

AU: Lithium-heparin tubes were used and this is added to line 106-107.

Line 99-101- You need to be more specific about blood, gastrointestinal and intestinal fluid sample point to make the meaning clear to the readers. For example; at 0 (pre-treatment), 5, 10, 15, 20, 25, 30, and 45 min and 36 h post-dosing. Which sampling times you have used for pharmacokinetic?

AU: This has been added to lines 109-111 and 137-139. All timepoints were included in the pharmacokinetic modeling. 

Line 113-114- "The calves received 2 mg/kg of flunixin meglumine intravenously prior to surgery and 24 hours after surgery". Could this drug have caused a change in the pharmacokinetic profile of ceftiofur? What impact this might have had on data interpretation, if any?

AU: Without undue additional study, we cannot determine the influence that flunixin may have on the pharmacokinetics of ceftiofur. We considered this, but decided not to entertain any speculation on a proposed effect. In addition, there is no evidence of drug interactions when ceftiofur and flunixin are administered in cattle (Gorden et al., J Vet Pharmacol Ther. 2018; 41 (1):76-82). This clarification and reference have been added to lines 124-125. 

Line 115 and 127- You need to be more specific about predetermined time points.

AU: This is now included in lines 137-139.

Line 150- Are you used the WinNonlin program for statistical difference. I don’t understand.

AU: This has been removed and a description of the statistical analysis of the PK parameters has been added to line 265-268.

Line 251- Did you compare the pharmacokinetic parameters statistically?

AU: This has been added to line 265-268. 

Results

Line 274- Pharmacokinetic modelling- Did you write your results here without any statistical comparison?

AU: Asterisks have been added to Table 1 to indicate differences based on statistical analysis. The results portion has been edited to reflect these differences. 

Discussion: Discussion of the study is not sufficient: the pharmacokinetic parameters were ignored. After the statistical analysis, I suggest re-evaluation of the relevant parts of the pharmacokinetic parameters.

AU: We thank the reviewer for this suggestion. We have added more to the discussion (primarily in lines 383-407) to satisfy the reviewer’s concern. 

Line 364-367- This line is like a repetition of the sentences in Line 43-44. I suggest you to combine these in introduction. 

AU: These have been condensed as suggested.

Since your hypothesis is especially about the efficacy of ceftiofur in intestinal flora, you can instead add articles about the use of ceftiofur in calves with diarrhea. For this reason, I suggest below articles could be added to the discussion section of this study.

1. Constable P. D. 2009. Treatment of calf diarrhea: antimicrobial and ancillary treatments. Vet. Clin. North Am. Food Anim. Pract. 25: 101–120.

2. Feray ALTAN, Kamil UNEY, Ayse ER, Gul CETIN, Burak DIK, Enver YAZAR, and Muammer ELMAS. Pharmacokinetics of ceftiofur in healthy and lipopolysaccharide-induced endotoxemic newborn calves treated with single and combined therapy. J Vet Med Sci. 2017 Jul; 79(7): 1245–1252.

AU: While there is certainly the possibility that studies like ours could be used to investigate the efficacy of various antimicrobials on enteric pathogens for the treatment of diarrheal disease, our focus is on the impact GI tract drug concentrations on normal enteric bacteria to understand the effect of formulation and dosing on selection AMR bacteria and disruption of the microbiome. As our study was not designed to assess treatment efficacy, we believe that extrapolating our findings to include this information would be an inappropriate over interpretation of our findings and unnecessarily confuse the readers. 

Line 367-368- This information is new and should be moved to the introduction section in line 54.

AU: This has been moved as suggested.

Line 370-372- This line is like a repetition of the sentences in Line 54-60. I suggest you to delete in discussion.

AU: These have been condensed as suggested.

Line 372-374- I think it would be more appropriate for you to discuss this sentence in a different way rather than repeating what you have written in introduction section.

AU: This has now been revised and is in lines 379-382.

Line 375-377- You can't tell these without any statistical analysis. Also the table you have presented is so confused that I cannot reach this conclusion by looking at your table.

AU: Thank you for this suggestion. This section has been edited to reflect the statistical analysis. We will modify the tables to attempt to make them less confusing. We have added asterisks to signify significant differences and corrected the ISF values. 

Line 376- Cmax of CCFA in ISF was not lower than CCFA. Please be carefully. The data you type in the table should be consistent with what you type in the text.

AU: You are correct. This error has been corrected in line 384. 

Line 379-384- You were stated “The ISF concentrations after CCFA injection were much higher. This likely occurred because of a longer time for equilibration between plasma and interstitial tissue fluid for CCFA. This also produced longer persistence of ceftiofur and its metabolites in intestinal fluids.” You have indicated in your table that the half-life of CCFA is shorter than CHLC in ISF. Can you explain why the half-life and the concentration of CHLC is long and low in ISF.

AU: We made an error in our calculation of the ISF parameters in our original submission. We apologize for this oversight. As we noted in response to Reviewer #1, this has been corrected in the tables. 

Line 385-409- Ceftiofur are beta-lactam antimicrobial drugs. Activity of beta-lactams is time-dependent kill characteristics, and the most useful and predictable parameter for optimal bactericidal activity is %T>MIC. Use this parameter to give information about the susceptible bacteria for ceftiofur. You can’t this say it through MIC 90 and concentration. You should determine to %T>MIC for susceptible bacteria.I suggest below article could be added

Turnidge, J. D. (1998). The pharmacodynamics of beta‐lactams. Clinical Infectious Diseases, 27, 10–12.

AU: Yes, we are familiar with Dr. Turnidge’s work. We can indeed calculate T>MIC for the wild-type distribution of MIC for E. coli. However, as we have discussed, because of low MIC values for E. coli, the T>MIC was sufficiently long during the dosing interval to achieve inhibition. 

Table 1. I think you need more explanation in this table. Why did you not compare the differences in plasma concentrations?

AU: Statistical comparisons between the two drugs in the various compartments have been added. 

Figure 5. Calves?????

AU: Each bar represents and individual calf. This has been added to the figure legend.

---

## [Decision Letter · Decision Letter 1]

3 Sep 2019

[EXSCINDED]

PONE-D-19-14785R1

Ceftiofur formulation differentially affects the intestinal drug concentration, resistance of fecal Escherichia coli, and the microbiome of steers

PLOS ONE

Dear Dr. Foster,

Thank you for submitting your manuscript to PLOS ONE. After careful consideration, we feel that it has merit but does not fully meet PLOS ONE’s publication criteria as it currently stands. Reviewer 2 indicates that important information is still missing from the pharmacokinetic analysis and that the differences in the results from parameters investigated in plasma needs to be discussed. Therefore, we invite you to submit a revised version of the manuscript that carefully addresses the points raised during the review process.

We would appreciate receiving your revised manuscript by Oct 18 2019 11:59PM. To enhance the reproducibility of your results, we recommend that if applicable you deposit your laboratory protocols in protocols.io, where a protocol can be assigned its own identifier (DOI) such that it can be cited independently in the future. For instructions see: http://journals.plos.org/plosone/s/submission-guidelines#loc-laboratory-protocols

We look forward to receiving your revised manuscript.

Kind regards,

Kristin Mühldorfer

Academic Editor

PLOS ONE

Reviewers' comments:

Reviewer's Responses to Questions

**Comments to the Author**

1. If the authors have adequately addressed your comments raised in a previous round of review and you feel that this manuscript is now acceptable for publication, you may indicate that here to bypass the “Comments to the Author” section, enter your conflict of interest statement in the “Confidential to Editor” section, and submit your "Accept" recommendation.

Reviewer #2: All comments have been addressed

2. Is the manuscript technically sound, and do the data support the conclusions?

Reviewer #2: Yes

3. Has the statistical analysis been performed appropriately and rigorously? 

Reviewer #2: Yes

4. Have the authors made all data underlying the findings in their manuscript fully available?

Reviewer #2: Yes

5. Is the manuscript presented in an intelligible fashion and written in standard English?

Reviewer #2: Yes

6. Review Comments to the Author

Reviewer #2: There are still a lack of information on pharmacokinetics. You need to be more specific about pharmacokinetic parameters as these could affect your outcome. You compared the differences in only plasma concentrations. Why did you not compare the differences in other pharmacokinetic parameters? The Tmax and AUC of CCFA in plasma were difference than CHCL in plasma. Additionally, the Cmax of CCFA in plasma is lower than of CHCL in plasma. The t1/2λz CCFA in plasma was longer than CHCL in plasma. Could you suggest some reasons?

7. PLOS authors have the option to publish the peer review history of their article (what does this mean?). If published, this will include your full peer review and any attached files.

Reviewer #2: No

---

## [Author Response · Author response to Decision Letter 1]

17 Sep 2019

Thank you to the editors and reviewers for their helpful comments. We have attempted to address all of the stated concerns. The line numbers listed here in our responses correspond to the version of the manuscript without tracked changes. 

Reviewer #2: There are still a lack of information on pharmacokinetics. You need to be more specific about pharmacokinetic parameters as these could affect your outcome. You compared the differences in only plasma concentrations. Why did you not compare the differences in other pharmacokinetic parameters? The Tmax and AUC of CCFA in plasma were difference than CHCL in plasma. Additionally, the Cmax of CCFA in plasma is lower than of CHCL in plasma. The t1/2λz CCFA in plasma was longer than CHCL in plasma. Could you suggest some reasons?

AU: Our apologies that this was not more clearly stated, but we have compared the PK parameters in all the different matrices and highlighted those in Table 1 that were significantly different. In lines 291-298, we have expanded our description the observed differences in the PK parameters and have added information in lines 387-390 along with a reference highlighting the impact of the slow-release formulation of CCFA.

---

## [Editor Report · Decision Letter 2]

20 Sep 2019

Ceftiofur formulation differentially affects the intestinal drug concentration, resistance of fecal Escherichia coli, and the microbiome of steers

PONE-D-19-14785R2

Dear Dr. Foster,

We are pleased to inform you that your manuscript has been judged scientifically suitable for publication and will be formally accepted for publication once it complies with all outstanding technical requirements.

Few additional Editor comments are listed below and within one week, you will receive an e-mail containing information on the amendments required prior to publication. When all required modifications have been addressed, you will receive a formal acceptance letter and your manuscript will proceed to our production department and be scheduled for publication.

With kind regards,

Kristin Mühldorfer

Academic Editor

PLOS ONE

Additional Editor Comments:

1) I would recommend to explain standard abbreviations (CCFA, CHCL, ISF, MIC) in tables and figures too, as readers might not always follow the manuscript at all. For example, the title could be extended accordingly.  

2) Check upper and lower case in titles and content from tables and figures. 

3) Check the "p" from p-values, upper or lower case? In table 3, the "p" is italicized. Should be uniform!

4) page 14, line 295: "... metabolites were 2-3 times greater ..."

5) page 14, line 298: replace "those" by "although" or similar

---

## [Editor Report · Acceptance letter]

25 Sep 2019

PONE-D-19-14785R2 

Ceftiofur formulation differentially affects the intestinal drug concentration, resistance of fecal *Escherichia coli*, and the microbiome of steers 

Dear Dr. Foster:

I am pleased to inform you that your manuscript has been deemed suitable for publication in PLOS ONE. Congratulations! Your manuscript is now with our production department. 

With kind regards,

on behalf of

Dr. Kristin Mühldorfer 

Academic Editor

PLOS ONE